# LLM AS GNN: GRAPH VOCABULARY LEARNING FOR GRAPH FOUNDATION MODEL

## ABSTRACT

Graphs typically exhibit distinctive structure and domain-specific knowledge, motivating the development of a Graph Foundation Model (GFM) capable of generalizing across various graphs and tasks. While recent efforts have focused on combining the strengths of Large Language Models (LLMs) and Graph Neural Networks (GNNs), they often struggle to maximize mutual benefit due to the decoupled architectures. Moreover, existing methods assign out-of-vocabulary (OOV) tokens to nodes, which are incompatible with the natural language vocabulary for task-oriented prompt generation, hindering knowledge transfer in GFM. In this paper, we introduce PromptGFM, a versatile GFM grounded in graph vocabulary learning, comprising two key components: (1) Graph Understanding Module, which explicitly replicates the finest GNN workflow in the language space using LLMs, enabling seamless GNN-LLM integration and elegant graph-text alignment; (2) Graph Inference Module, where we establish a novel language-based graph vocabulary to ensure expressiveness, transferability, and scalability. This vocabulary enables the generation of readable instructions for LLM inference, resolving modality incompatibility and facilitating positive transfer. Extensive experiments demonstrate the superiority of PromptGFM in node classification and link prediction, along with its strong transferability across different datasets and tasks. The code is available at https://anonymous.4open.science/r/PromptGFM.

## 1 INTRODUCTION

Graphs, representing intricate and complex relationships between nodes, are ubiquitous across various real-world domains, including citation networks (Eto, 2019; Hu et al., 2020; Buneman et al., 2021), social networks (Kempe et al., 2003; Myers et al., 2014), and molecular graphs (Wieder et al., 2020; Jin et al., 2024). These graphs typically exhibit distinctive non-Euclidean data structures while embody essential domain-specific knowledge. In this context, most graph learning approaches necessitate individual training and deployment tailored to specific datasets or tasks. To overcome this limitation, we aim to build a Graph Foundation Model (GFM) capable of generalizing across different graphs and tasks (Mao et al., 2024; Xia et al., 2024).

With the advent of Large Language Models (LLMs), significant efforts have been dedicated to harnessing their powerful understanding and inference capabilities alongside traditional Graph Neural Networks (GNNs) to tackle broad challenges in graph machine learning. As Figure 1 shows, existing

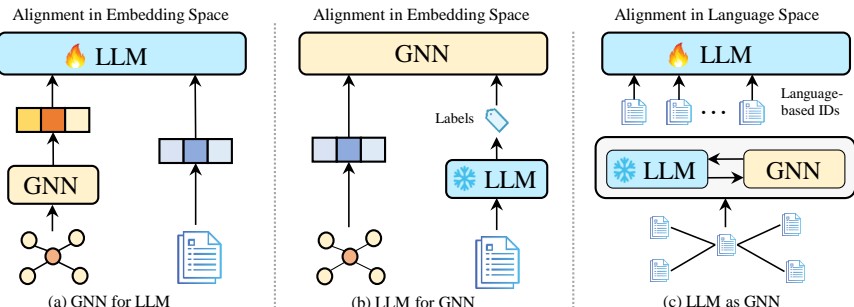

Figure 1: Overview of different GNN-LLM integration methods for graph-text alignment: (a) **GNN for LLM** and (b) **LLM for GNN** use decoupled architectures in the embedding space. (c) In this work, we aim to function **LLM as GNN** to achieve alignment directly in the language space.

studies focus on modeling text-attributed graphs, which can be summarized as follows: **(1) GNN for LLM.** GNNs produces structure-aware embeddings that enhance original textual embeddings, boosting LLM inference (Tang et al., 2024; Chai et al., 2023; Liu et al., 2024b). **(2) LLM for GNN.** LLMs contribute additional node features or labels derived from textual data, supervising the training of GNN predictions (Chen et al., 2024c; Liu et al., 2024a; Zhu et al., 2024). Nonetheless, concurrent loosely decoupled architectures struggle to maximize the advantages of both GNNs and LLMs simultaneously, resulting in suboptimal graph-text alignment.

Recently, a noteworthy trend has emerged toward implementing **LLM as GNN**, where LLMs function as GNNs to capture graph semantics and structure. A few studies design structure verbalizers to convert graph structures into code-like or heuristic prompts, enabling LLMs to comprehend and encode the graph through text (Ye et al., 2024; Wang et al., 2024; Chen et al., 2024a). However, we argue there are currently no genuine instances of this category. As Figure 2(a) shows, the essence of a true GNN lies in its message-passing paradigm, where each layer involves key components such as neighbor sampling, aggregation, update, and optimization (Kipf & Welling, 2017; Velickovic et al., 2017). By stacking multiple GNN layers, structure-less embeddings are gradually transformed into structure-rich embeddings, capturing higher-order signals. Motivated by the absence of essential properties of GNNs, a critical challenge arises: *Can we leverage LLMs to faithfully replicate GNNs to capture both graph semantics and structures simultaneously?*

Even worse, existing models are all confined to a shared embedding space. They intuitively assign each node in the graph as an out-of-vocabulary (OOV) token, relying on ID-based embeddings for downstream graph-specific tasks (Tang et al., 2024; Ye et al., 2024). Unfortunately, these graph embeddings are inherently incompatible with language-based token embeddings due to mismatch vocabularies, leading to semantic discrepancies when constructing natural language instructions for LLM inference. More importantly, this incompatibility makes it challenging for this graph-specific knowledge to transfer or scale to other graphs and tasks. To achieve positive transfer, an urgent challenge emerges: *Can we replace the OOV tokens with compatible node representations to build versatile graph foundation model?*

This paper aims to build a versatile GFM grounded in **graph vocabulary learning** (Mao et al., 2024; Cai, 2024). An ideal graph vocabulary should share following properties: **(1) Expressiveness:** the vocabulary encapsulates semantic and structural knowledge across various domain-specific graphs. **(2) Transferability:** each node in any graph is represented by one or more fundamental units within this vocabulary. **(3) Scalability:** the vocabulary is sufficiently inclusive to accommodate unseen nodes, even those from outside existing graphs. Since natural language is a highly expressive medium made up of meaningful and transferable tokens (Raffel et al., 2020; Radford et al., 2021; Palo et al., 2023), we establish a universal graph vocabulary within the language space to develop a versatile GFM, called PromptGFM. In particular, PromptGFM consists of two key components:

**Graph Understanding Module.** To function LLMs as GNNs, we prompt LLMs to explicitly replicate the core principles of GNNs within the language space. As shown in Figure 2, we use textual attributes as initial features and meticulously design a series of prompts to align with the GNN workflow at the finest granularity. Specifically, we sample one-hop neighbors for each node to convey graph structure, and then use straightforward prompts to simulate a more flexible aggregation-update mechanism. Additionally, we design heuristic prompts to reflect the key idea of contrastive loss, as analogy to mean pooling in unsupervised graph learning (Hamilton et al., 2017). By multiple rounds of LLM calls, we faithfully reproduce the iterative message-passing paradigm of GNNs, progressively refining verbose textual attributes into concise but meaningful textual representations, rather than relying on numerical embeddings. Furthermore,

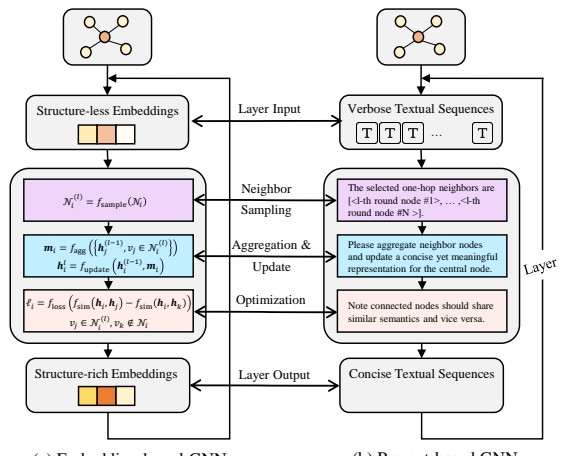

Figure 2: The LLM-driven replication of the GNN workflow to refine textual representations and capture high-order signals. We achieve fine-grained alignment between traditional embedding-based GNN and our prompt-based GNN.

our prompt-based GNN successfully inherits the advantages of embedding-based GNNs, preserving critical node semantics while capturing higher-order relationships within the graph. In essence, LLMs function as GNNs, and GNNs can be viewed as LLMs, unleashing the potential of GNN-LLM integration and empowering elegant graph-text alignment.

**Graph Inference Module.** Since we have captured semantic and structural information through the prompt-based GNN, we propose decoupling these textual representations to establish a universal graph vocabulary, where each node is mapped to a finite sequence of tokens, essentially as language-based IDs. This vocabulary is universally transferable and scalable across diverse graphs, which resolves the incompatibility typically associated with OOV tokens. Building on this insight, we feed these language-based IDs to generate readable and coherent instructions composed entirely of transferable natural language tokens. Within a multi-instruction fine-tuning framework, we gather massive instructions from various graphs and tasks to effectively fine-tune an LLM, enabling it to transfer cross-graph and cross-task knowledge for handling unseen graphs and tasks. In conclusion, this universal graph vocabulary empowers us to thoroughly overcome the incompatibility challenge and build a general graph foundation model. The contributions are summarized as follows:

• We propose a graph foundation model capable of generalizing across all the graphs and tasks.

• We highlight the potential of functioning LLMs as GNNs, and propose prompting LLMs to drive a fine-grained replication of GNN flows that captures high-order signals within the language space. This approach facilitates a seamless GNN-LLM integration and achieves elegant graph-text alignment.

• We establish a universal graph vocabulary within the language space that resolves graph-text incompatibility. By multi-prompt instruction fine-tuning, its inherent transferability and scalability empower the acquisition of open-world global knowledge for a graph foundation model.

• We conduct extensive experiments on several public benchmarking datasets, demonstrating the effectiveness of our proposed PromptGFM framework in node classification and link prediction tasks, as well as its strong zero-shot transferability across various datasets and tasks.

## 2 RELATED WORKS

**GNN-LLM Integration.** LLMs have unlocked unprecedented potential for graph machine learning, inspiring the integration between GNNs and LLMs. **(1) GNN for LLM.** A common approach employs graph encoders to capture graph structure, aiding LLMs in comprehensive graph understanding. Many models, like GraphGPT (Tang et al., 2024) and GraphLLM (Chai et al., 2023), and GraphPrompter (Liu et al., 2024b), use GNNs or graph transformers as structure tokenizers, enabling synergistic fine-tuning with LLMs. Despite their prevalence, challenges persist in effectively coordinating the architecture and co-training of GNNs and LLMs. **(2) LLM for GNN.** Another approach harnesses LLMs to provide additional labels and features for GNN. In practice, LLM-GNN (Chen et al., 2024c), GraphEdit(Guo et al., 2024), and OpenGraph (Xia et al., 2024) leverage LLMs to generate node-level and edge-level labels, addressing data sparsity issues. **OFA** (Liu et al., 2024a), **ENGINE** (Zhu et al., 2024), and TAPE (He et al., 2024) further direct LLMs to produce additional features and explanations to overcome semantic deficiencies. Nevertheless, a notable drawback lies in the heavy reliance on the instructions design and the quality of LLM-generated content, which inevitably introduces noise and negatively impacts performance. **(3) LLM as GNN.** In this paradigm, LLMs themselves are designed to function as GNNs. Existing efforts, such as LLaGA (Chen et al., 2024a), InstructGLM (Ye et al., 2024), and InstructGraph (Wang et al., 2024), employ structure verbalizers to convert graph structures into code-like or heuristic prompts for LLM inference. However, they struggle to capture high-order connections without a true GNN mechanism, thus not fully functioning LLMs as GNNs. Overall, the current decoupled integration of LLMs and GNNs fails to fully exploit the strengths of both architectures. This limitation motivates us to propose a new paradigm where LLMs function as GNNs at every level, maximizing their synergistic potential.

**Graph-Text Alignment in Embedding Space.** Existing graph-text alignment preserves features from different modalities, while facilitates their coordination in the embedding space. With graph encoders as prefixes, some studies actively align graph-aware embeddings with LLMs and finetune them alongside language-based embeddings, such as G-Prompt (Huang et al., 2023) and GraphAdapter (Huang et al., 2024). An alternative way is to derive distinctive features via a two-tower architecture and apply alignment techniques for mutual benefits, such as contrastive learning (Li et al., 2023a;

Brannon et al., 2023; Tang et al., 2024), iterative training (Zhao et al., 2023a; Zhu et al., 2024), and knowledge distillation (Mavromatis et al., 2023). Nevertheless, these strategies bridge modality gaps by mapping data to a shared embedding space, but they face challenges in terms of flexibility, transferability, and scalability when dealing with natural language tokens from task-specific templates.

**Graphs Foundation Models.** A GFM aims to enable transferability across different datasets and tasks (Liu et al., 2023). Observations have shown the key challenge lies in finding a graph vocabulary, identifying transferable units to encode invariance on graphs (Mao et al., 2024). Specifically, GraphGPT (Tang et al., 2024) assumes a unique ID for each node, creating a dataset-specific vocabulary. MoleBERT (Xia et al., 2023) defines a molecular graph vocabulary by converting atomic properties into chemically meaningful codes. Despite their success, ID-based vocabularies are typically domain-specific, lacking in-context learning capabilities and exhibiting poor transferability across different domains.. Recently, significant efforts (Fatemi et al., 2024; Wang et al., 2023; Zhao et al., 2023b; Liu et al., 2024a) have been dedicated to effectively understanding and inferring graphs in a natural language format, but none of them attempts to extend a language-based graph vocabulary to fully leverage the inherent transferability of natural language. In this paper, we aim to build an expressive graph vocabulary using natural language tokens, which means to represent each node with a finite sequence of language tokens. Our work thoroughly resolves the semantic discrepancy between graph and text, advancing the development of a versatile GFM.

## 3 PRELIMINARIES

**Graph Data.** A graph is formally represented as $G = (V, E, X)$, where $V$ is the set of nodes, $E$ is the set of edges. In this work, each node $v_i \in V$ is associated with a textual description $X_i = \left(x_i^1, x_i^2, \ldots, x_i^{n_i}\right)$, where each $x_i^k \in \mathcal{X}, k = 1, \ldots, n_i$. Here, $X$ denotes the textual attributes for nodes, and $\mathcal{X}$ represents the natural language token dictionary.

**Graph Neural Networks.** GNNs have gained widespread recognition as state-of-the-art models in graph machine learning, with most operating through a message passing paradigm (Wu et al., 2021). In this framework, a GNN begins by selecting neighboring nodes to the target node, then aggregates their representations to capture the local structure of the graph. The target node subsequently updates its own representation using the aggregated information. Mathematically, for a given node $v_i$, the $l$-th layer of a general GNN is formulated as:

$$
\begin{aligned}
\mathcal{N}_i^{(l)} &= f_{\text{sample}} \left( \mathcal{N}_i \right), \\
\mathbf{m}_i^{(l)} &= f_{\text{agg}} \left( \left\{ \mathbf{h}_j^{(l-1)}, v_j \in \mathcal{N}_i^{(l)} \right\} \right), \\
\mathbf{h}_i^{(l)} &= f_{\text{update}} \left( \mathbf{h}_i^{(l-1)}, \mathbf{m}_i^{(l)} \right)
\end{aligned}
\tag{1}
$$

where $\mathbf{h}_i^{(l)}$ is the embedding of node $v_i$ in the $l$-th layer. $\mathcal{N}_i$ is the full set of its neighbors and $\mathcal{N}_i^{(l)}$ represents sampled neighbors in the $l$-th layer. To capture high-order relationships, we stack multiple GNN layers and acquire final embeddings by adopting the last layer embedding $\mathbf{h}_i = \mathbf{h}_i^{(L)}$ or mean pooling $\mathbf{h}_i = f_{\text{mean}} \left( \mathbf{h}_i^{(1)}, \ldots, \mathbf{h}_i^{(L)} \right)$, where $L$ is the number of layers (Grattarola et al., 2024). For GNN training, a contrastive loss with negative sampling is commonly used in unsupervised graph learning (Hamilton et al., 2017; Velickovic et al., 2019). The goal is to increase the similarity between connected nodes and decrease it between unconnected nodes:

$$
\ell = f_{\text{loss}} \left( f_{\text{sim}} \left( \mathbf{h}_i, \mathbf{h}_j \right), f_{\text{sim}} \left( \mathbf{h}_i, \mathbf{h}_k \right) \right), v_j \in \mathcal{N}_i^{(l)}, v_k \notin \mathcal{N}_i
\tag{2}
$$

where $v_j$ is a positive sample and $v_k$ is a negative one. $f_{\text{sim}} \left( \cdot \right)$ measures the similarity between two node embeddings, such as dot product or cosine similarity. $f_{\text{loss}} \left( \cdot \right)$ represents the contrastive loss, which could be margin-based or cross-entropy based loss function.

## 4 METHODOLOGY

In this section, we describe the pipeline for our proposed PromptGFM framework. We highlight the component-wise reproduction of the GNN framework in graph understanding module and our language-based graph vocabulary in graph inference module, as illustrated in Figure 3 and Figure 4.

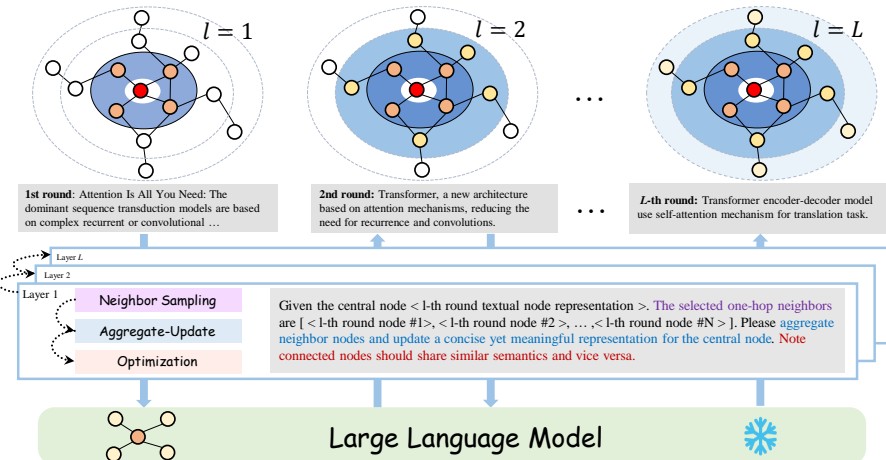

Figure 3: Graph understanding module. We prompt LLMs to achieve fine-grained reproduction of traditional GNN workflow, refining verbose textual representations into concise yet meaningful ones. In the prompt, neighbor sampling (see Equation 3) is highlighted in purple, the aggregation-update mechanism (see Equation 4) in blue, and the optimization in red.

## 4.1 GRAPH UNDERSTANDING MODULE

The graph understanding module aims to generate expressive representations for each node within the graph, supporting the subsequent graph inference module. The key to this task lies in effectively capturing and aligning the semantic and structural information, where LLMs and GNNs offer distinct advantages (Li et al., 2023b; Ren et al., 2024). However, the decoupled nature between GNNs and LLMs leads to potential semantic discrepancy and information loss. In this module, we propose a prompt-based GNN by functioning LLM as GNN, where core GNN operations are faithfully replicated within natural language space using LLMs.

**GNN Replication with LLMs.** Our priority is to design appropriate prompts that reflect the GNN workflow and guide LLMs to execute them. This requires considering three aspects: **(1) Graph Representation:** How can we effectively convey the node features and local structure to LLMs? **(2) Graph Structure:** How can we achieve message passing to capture the global structure? **(3) Graph Semantics:** How can we refine the core semantics to produce concise yet meaningful representations?

As shown in Figure 3, we conduct a fine-grained replication of GNN within the language space, i.e. prompt-based GNN. First, we use the LLM to summarize raw textual attributes as input to the GNN, akin to node initialization with low-dimensional embeddings. Then, we follow the general workflow of a GNN layer to inform prompt engineering. Due to the prompt length limitation, we sample its one-hop neighbors and extract their corresponding textual summaries as follows:

$$\left\{ X_j^{(l-1)}, \{v_j\} \subset \mathcal{N}_i \right\} \leftarrow \text{Prompt}_{\text{sample}} \left( X^{(l-1)}, \mathcal{N}_i \right), \quad (3)$$

where $X^{(l-1)} = \left\{ X_0^{(l-1)}, X_1^{(l-1)}, \ldots, X_{|V|-1}^{(l-1)} \right\}$ denotes the textual representations of all nodes at the previous layer, and $\left\{ X_j^{(l-1)} \right\}$ corresponds to the selected neighbors from $\mathcal{N}_i$. $\text{Prompt}_{\text{sample}}(\cdot)$ refers to sampling a subset of one-hop neighbors and obtaining their textual representations, similar to how traditional GNNs reduce computational overload. Prompts can be found in Appendix C. Leveraging this context, we can directly use natural language prompts to guide the most essential message aggregation-update process, which can be formulated as:

$$X_i^{(l)} \leftarrow \text{Prompt}_{\text{agg-update}} \left( \left\{ X_j^{(l-1)}, \{v_j\} \subset \mathcal{N}_i \right\}, X_i^{(l-1)} \right). \quad (4)$$

where $\text{Prompt}_{\text{agg-update}}(\cdot)$ is a prompt that aggregates the neighborhood information with previous representations and updates the node $v_i$ to produce $X_i^{(l)}$ at the current layer. Unlike traditional operators like mean and weighted aggregators, this prompt-based method allows for more flexible message passing without constraints. To handle diverse downstream tasks, we consider an unsupervised graph learning setting that commonly leverages contrastive learning for optimization, encouraging semantic similarity between neighboring nodes and dissimilarity between distant ones. We rely on prompts to intuitively steer this process because negative sampling is redundant in the situation.

After repeating $L$ rounds for all nodes, we obtain the final textual representation $T_i = X_i^{(L)}$ for each node $v_i \in V$. These textual representations are rich in both semantic and structural information, effectively solving the outlined issues: **(1) Graph Representations:** Through multi-layer propagation, one-hop neighbor descriptions are equivalent to an adjacency matrix, thereby representing the entire graph structure. **(2) Graph Structure:** To capture high-order relationships, we repeatedly call the LLM with the same prompts, with the output of each round serving as the input for the next. **(3) Graph Semantics:** Simultaneously, we instruct the LLM to produce concise yet meaningful textual representations for each node, gradually refining for denser and richer semantics.

**Embedding-based GNNs vs. Prompt-based GNNs.** We systematically compare existing embedding-based GNNs with our proposed prompt-based GNN and highlight our advantages. Overall, we empower LLM to faithfully mirror the whole GNN workflow within language space.

• **Input and output.** In the embedding-based GNN framework, for each node, *structure-less embeddings* are progressively refined to *structure-rich embeddings*, whereas *verbose textual sequences* are gradually converted to *concise textual sequences* in our prompt-based GNN.

• **Message passing.** The *multi-layer embedding updates* are mirrored by *multi-round LLM calls* in the language space, both of which progressively refining the representations in different spaces.

• **Neighbor sampling.** The *neighbor sampling* operation used to reduce computational load in traditional embedding-based GNNs is analogous to the *selected one-hop neighbor descriptions* employed to address prompt length limitations in prompt-based GNNs.

• **Aggregation-update mechanism.** Embedding-based GNNs use *predefined operators* (e.g., mean aggregator, weighted aggregator, or LSTM aggregator) to achieve message passing in the embedding space, while prompt-based GNNs use *straightforward prompts* to guide LLMs in executing the process more flexibly without predefined rules.

• **Optimization.** In prompt-based GNNs, we use heuristic prompts at each layer to reflect the key idea of contrastive loss. These *cumulative layer-by-layer prompts* are comparable to the layer-wise loss combination, formally as *mean pooling*, commonly seen in embedding-based GNNs.

Our graph understanding module relies solely on prompting LLMs, using text as both input and output, without requiring fine-tuning of LLMs. In summary, our component-wise replication of GNNs serves as an example of functioning LLMs as GNNs, effectively achieving seamless GNN-LLM integration and elegant graph-text alignment.

## 4.2 GRAPH INFERENCE MODULE

The primary role of the graph inference module in our PromptGFM is to acquire transferable open-world knowledge through instruction fine-tuning with LLMs, making it adaptable to different graphs and tasks. Current methods treat nodes as OOV tokens and merge them with task-oriented templates to form instructions. However, distinct vocabularies across modalities create incompatibility between graph and language-based embeddings, leading to semantic discrepancies and limiting the transfer of graph-specific knowledge. In response, we propose a novel language-based graph vocabulary that resolves this incompatibility and enables readable and coherent instructions for LLM inference.

**Graph Vocabulary Learning.** Learning a transferable graph vocabulary, whose fundamental units can represent each node, plays a central role in building GFMs. The effectiveness of a graph vocabulary is determined by three essential criteria: expressiveness, transferability, and scalability. Since each node has been associated with a textual representation that encapsulates its core semantics and local structure, we intuitively propose establishing a graph vocabulary within the natural language space using these rich representations. To this end, we introduce an expressive and universal language-based graph vocabulary, where each node is represented by such a finite sequence of language tokens, i.e. a language-based ID. Formally, our graph vocabulary can be defined as follows:

$$\mathcal{F} : V \to T, \tag{5}$$

where $T = \left\{ T_0, T_1, \ldots, T_{|V|-1} \right\}$ denotes language-based IDs of all nodes in $V$. Each node $v_i$ is mapped to a sequence $T_i = \left( t_i^1, t_i^2, \ldots, t_i^{m_i} \right)$, where $t_i^k \in \mathcal{X}$, and $\mathcal{X}$ is natural language token dictionary. Evidently, language tokens serve as the fundamental units in our graph vocabulary, with structured sequences representing nodes in the graph, akin to words in a conventional vocabulary.

Figure 4: An instance of graph inference module in link prediction, where language-based IDs are indexed from the graph vocabulary to generate readable instructions using task-oriented templates. We adopt multi-instruction fine-tuning framework to learn transferable global knowledge for a GFM.

Our graph vocabulary satisfies all expected criteria: **(1) Expressiveness.** The textual representations in the vocabulary have captured rich semantic and structural information from an open-world setting. **(2) Transferability.** Like natural language vocabulary, our graph vocabulary also shares natural language tokens as fundamental units, enabling direct transfer across different graphs and tasks. **(3) Scalability.** Any node, whether previously seen or not, can be comparable to existing nodes via its language-based ID, effectively resolving the semantic discrepancy of OOV tokens.

**Instruction Fine-Tuning.** We employ a multi-instruction fine-tuning framework for LLM inference to incorporate various graphs and tasks (Chung et al., 2024; Wei et al., 2022). As illustrated in the Figure 4, we index nodes from the graph vocabulary and integrate their language-based nodes into task-oriented prompt templates to form comprehensive instructions:

$$\mathcal{T} \leftarrow \text{Prompt}_{\text{template}} \left( T, G \mid \mathcal{F} \right), \tag{6}$$

where $\text{Prompt}_{\text{template}} \left( \cdot \right)$ concatenates language-based IDs $T$ with appropriate templates to generate a collection of completed instructions $\mathcal{T}$. Notably, these instructions are fully readable and composed entirely of natural language tokens. In light of LLMs' strengths in understanding and generating text, we effectively convert all question-answering tasks into a unified text-to-text format (Mishra et al., 2022). Let $Y$ denote target output sequences, the loss function for LLM fine-tuning is computed as:

$$\mathcal{L} = -\sum_{j=1}^{|Y|} \log \Pr \left( Y_j \mid \mathcal{T}, Y_{<j} \right), \tag{7}$$

where $\Pr \left( Y_j \mid \mathcal{T}, Y_{<j} \right)$ is the probability of the $j$-th token $Y_j$ in the output sequence $Y$, conditioned on the instruction $\mathcal{T}$ and all previous tokens $Y_{<j} = (Y_1, Y_2, \ldots, Y_{j-1})$. This probability is computed by the LLM in an autoregressive manner, following the standard next-token prediction approach used in models like T5 (Raffel et al., 2020), FLAN (Wei et al., 2022), and LLaMA (Touvron et al., 2023).

**Constrained Decoding with Prefix Tree Search.** Generating candidate neighbors in link prediction may cause LLM hallucination issues. In response, we introduce a constrained decoding method using a prefix tree search strategy to regulate LLM outputs (Cao et al., 2021; Tan et al., 2024). Specifically, we collect the language-based IDs of all candidate nodes and craft a prefix tree, where each tree node equals to a natural language token. Each unique path from the root to a leaf corresponds to the language-based ID of a node. During autoregressive generation, each new token is constrained by the previous tokens to follow a valid path in the prefix tree. This ensures that the prediction is associated with an actual graph nodes, effectively preventing hallucinations. This success is largely due to the discrete language-based IDs, further demonstrating the flexibility of the proposed graph vocabulary.

**Generalization of Graph Foundation Model.** We employ prompt-based GNN to directly propagate textual representations within a graph, capturing semantic and structural information. By decoupling these nodes from their original graphs, we establish a universal graph vocabulary. To develop a versatile GFM, we can extract nodes and generate instructions from different graphs. Thus, we can co-train across graphs and tasks by fine-tuning a unified LLM. This approach enables the acquisition of open-world global knowledge and inclusive accommodation of any unseen graphs or tasks.

## 5 EXPERIMENTS

In this section, we conduct extensive experiments to address the following questions: **RQ1:** How does PromptGFM perform on supervised node classification and link prediction? **RQ2:** Can it generalize to unseen graphs and tasks as a versatile GFM? **RQ3:** How does each module contribute to the overall performance? **RQ4:** What affect the GNN replication, and what insights can be gained from it?

Table 1: Comparison with four categories of baselines on node classification accuracy. Results of GNN-LLM models are sourced from original papers or this study (Chen et al., 2024b) without standard deviation. 'N/A' denotes unreported metrics. Macro-F1 scores are in Appendix E.

| Model | Cora | Citeseer | PubMed | ogbn-arxiv |
|---|---|---|---|---|
| MF | 60.07±2.69 | 62.33±2.08 | 59.82±1.42 | 68.47±0.92 |
| MLP | 62.29±4.69 | 64.42±2.43 | 62.88±1.88 | 62.07±0.35 |
| GCN | 82.47±3.89 | 76.11±3.34 | 77.36±1.07 | 66.15±0.46 |
| GAT | 82.92±2.58 | 77.30±2.57 | 74.36±3.48 | 65.29±0.64 |
| SAGE | 83.69±2.43 | 73.17±3.83 | 83.22±1.86 | 68.78±0.77 |
| RevGNN | 86.90±1.72 | 77.34±2.59 | 82.16±2.27 | 70.43±0.38 |
| AGNN | 77.64±2.55 | 73.14±1.93 | 73.55±0.62 | 60.63±0.42 |
| DNA | 80.81±3.38 | 73.64±2.98 | 80.68±1.33 | 58.51±0.67 |
| SGFormer | 82.36±2.88 | 73.76±3.03 | 78.92±1.63 | 63.44±0.95 |
| NodeFormer | 81.55±3.01 | 72.98±2.10 | 76.49±1.91 | 73.21±0.41 |
| OFA | 79.41 | 81.35 | N/A | 73.75 |
| LLaGA | 81.25 | 68.80 | N/A | 76.05 |
| ENGINE | 91.48 | 78.46 | N/A | 76.02 |
| GraphPrompter | 80.26 | 73.61 | **94.80** | 79.54 |
| PromptGFM | **91.72±1.06** | **84.49±1.37** | 90.67±1.16 | **80.58±0.54** |

Table 2: Comparison of discriminative link prediction performance across all datasets. Among the GNN-LLM models, only GraphPrompter and PromptGFM are applicable.

| Model | Cora | Citeseer | PubMed | ogbn-arxiv |
|---|---|---|---|---|
| MF | 66.43±1.13 | 70.12±1.99 | 59.34±1.03 | 60.26±0.92 |
| MLP | 70.11±2.50 | 74.76±1.80 | 62.88±1.88 | 72.64±3.37 |
| GCN | 77.15±2.20 | 78.72±2.11 | 77.36±1.07 | 80.89±1.66 |
| GAT | 70.44±3.80 | 77.17±3.30 | 74.36±3.48 | 76.25±1.90 |
| SAGE | 85.31±3.34 | 87.15±1.83 | 83.22±1.86 | 80.76±2.77 |
| RevGNN | 70.13±1.72 | 78.26±3.08 | 82.16±2.27 | 69.67±0.80 |
| AGNN | 71.52±2.62 | 74.23±1.90 | 73.55±0.62 | 75.56±0.67 |
| DNA | 62.76±1.85 | 63.96±1.07 | 58.56±3.36 | 69.13±2.51 |
| GraphPrompter | 90.10 | 91.67 | 86.49 | 73.21 |
| PromptGFM | **90.57±1.26** | **92.03±2.74** | **87.64±1.98** | **89.82±2.54** |

**Data description.** We introduce four public benchmarking datasets: Cora (McCallum et al., 2000), Citeseer (Giles et al., 1998), PubMed (Sen et al., 2008), and ogbn-arxiv (Hu et al., 2020), which are academic networks from different domains. Details can be found in Appendix A.

**Baselines.** We make comprehensive comparisons with existing methods across four categories: **(1) Graph-agnostic methods.** We consider basic models such as MF and MLP, which do not utilize graph structure. **(2) GNN-based methods.** We employ three fundamental GNN models: **GCN** (Kipf & Welling, 2017), **GAT** (Velickovic et al., 2017), and **GraphSAGE** (Hamilton et al., 2017). We also compare three different architectures: **ReVGNN** (Li et al., 2021), **AGNN** (Thekumparampil et al., 2018), and **DNA** (Fey, 2019). **(3) Transformer-based methods.** We explore **SGFormer** (Wu et al., 2023) and **NodeFormer** (Wu et al., 2022), which leverage transformer architectures to model graph data. **(4) GNN-LLM Integration methods.** Following the aforementioned taxonomy, we select **GraphPrompter** (Liu et al., 2024b) as an instance of using GNNs to enhance LLMs. **OFA** (Liu et al., 2024a) and **ENGINE** (Zhu et al., 2024) are examples of leveraging LLMs for GNNs. Besides, we incorporate **LLaGA** (Chen et al., 2024a) as an attempt of implementing LLM as GNN. We provide the details of these baselines in Appendix B.

**Reproduction Settings.** We implement our PromptGFM in PyTorch and run experiments on four NVIDIA RTX A6000 GPUs. The graph understanding module leverages OpenAI's GPT-3.5 while we fine-tune a T5 model (Raffel et al., 2020) in the graph inference module. We employ 10-fold cross-validation and report average results with standard deviation across all folds. For evaluation metrics, we use accuracy and Macro-F1 for node classification, and accuracy and HR@1 for discriminative and generative link prediction, respectively. We utilize the textual features to initialize the node embeddings for all embedding-based models. We provide further details in Appendix D.

## 5.1 OVERALL PERFORMANCE

**Node Classification and Link Prediction.** We train PromptGFM on a single graph from scratch and provide a comprehensive comparison in node classification and link prediction, shown in Tables

Table 3: Intra-domain cross-graph transferability on node classification. w/o training is direct inference using off-the-shelf LLM. supervised means training PromptGFM on a single graph from scratch through supervised learning.

| Setting | Acc | Macro-F1 |
|---|---|---|
| w/o training | 27.64 | 17.10 |
| Cora→Citeseer | 51.63 | 45.10 |
| arxiv→Citeseer | 60.34 | 54.81 |
| Cora+arxiv→Citeseer | 61.25 | 55.66 |
| supervised | 84.97 | 80.13 |

Table 4: Cross-task performance from link prediction (LP) to node classification (NC).

| Dataset | Setting | Acc | Macro-F1 |
|---|---|---|---|
| Cora | w/o training | 18.54 | 12.16 |
| | LP→NC | 60.74 | 55.42 |
| | supervised | 91.72 | 90.06 |
| Citeseer | w/o training | 27.64 | 17.1 |
| | LP→NC | 50.12 | 44.68 |
| | supervised | 84.49 | 80.13 |
| PubMed | w/o training | 39.12 | 39.84 |
| | LP→NC | 57.42 | 58.79 |
| | supervised | 90.67 | 91.82 |

Table 5: Inter-domain cross-graph transferability on node classification. We transfer knowledge from computer science domain (Cora, Citeseer, and arxiv) to biomedical domain (PubMed).

| Setting | Acc | Macro-F1 |
|---|---|---|
| w/o training | 39.12 | 39.84 |
| Cora→PubMed | 51.76 | 52.64 |
| Citeseer→PubMed | 40.12 | 42.38 |
| arxiv→PubMed | 60.21 | 62.02 |
| Cora+Citeseer→PubMed | 50.17 | 51.74 |
| Cora+arxiv→PubMed | 57.28 | 59.71 |
| Citeseer+arxiv→PubMed | 55.34 | 57.11 |
| Cora+Citeseer+arxiv→PubMed | 53.07 | 54.90 |
| supervised | 90.67 | 91.82 |

Table 6: Comparison of HR@1 for generative link prediction. Our PromptGFM is the only applicable model within GNN-LLM research.

| Model | Cora | Citeseer | PubMed |
|---|---|---|---|
| GCN | 5.95 | 6.82 | 0.51 |
| GAT | 2.22 | 3.59 | 0.28 |
| SAGE | 6.59 | 8.73 | 0.45 |
| PromptGFM | **8.21** | **8.90** | **1.21** |

1 and 2. PromptGFM demonstrates significant improvements over state-of-the-art models, with the following key insights: (1) Graph-based models generally outperform graph-agnostic methods, showing the value of graph structure. (2) In node classification, PromptGFM outperforms OFA, GraphPrompter, and ENGINE, which represent the two main approaches (GNN for LLM and LLM for GNN), which can be attributed to their decoupled integration. (3) PromptGFM outperforms LLaGA, which employs templates to convey graph structure for LLM inference. It is evident these heuristic prompts fail to capture sufficient high-order signals without a true GNN mechanism. Conversely, PromptGFM showcases the potential of using LLMs as GNNs via a prompt-based GNN, suggesting a new paradigm for LLM-GNN integration and the advancement of GFMs.

**Generative Link Prediction.** We conduct link prediction in a generative setting on Cora, Citeseer, and PubMed. Specifically, we split the graph data by links and construct an input graph using the training set. Given a specific node, we follow the transductive setting by predicting its unseen connections in the test set, where any node in the input graph can be a candidate. As Table 6 shows, our method consistently outperforms these traditional GNN models. Unfortunately, existing GNN-LLM works generally ignore this setting. This occurs because LLM outputs cannot map to the OOV token embeddings of specific nodes, resulting in unsolvable LLM hallucination issues. Our success lies in representing nodes as finite token sequences, enabling constrained decoding through prefix tree search to guide LLM outputs. This further highlights indispensable value of our graph vocabulary and generative capability of PromptGFM.

## 5.2 MODEL GENERALIZATION

**Cross-graph Generalization.** We evaluate zero-shot transferability across graphs, focusing on both intra-domain and inter-domain scenarios. Cora, Citeseer, and arxiv belong to the computer science domain as citation networks, and PubMed represents the biomedical domain. Table 3 shows the intra-domain results for node classification. First, all transfer results consistently outperform the w/o training variant by a large margin, highlighting the strong transferability of PromptGFM by effectively learning from other graphs. Not surprisingly, zero-shot results are inferior to graph-specific supervised learning from scratch. Fortunately, we observe improvements when co-training PromptGFM with Cora and Citeseer. The finding implies the potential to collect a large amount of graph data and train a highly comprehensive and knowlegable GFM in the future. Besides, Table 5 outlines cross-domain transfer from computer science to the biomedical field. Similarly, even in cross-domain settings, all zero-shot variants maintain superiority over the w/o training variant. Additionally, learning from scratch still performs best. However, unlike inter-domain scenarios, incorporating more source graph data does not always enhance performance in the target domain, possibly due to catastrophic forgetting or incompatible hyperparameters (Chen et al., 2024b).

Figure 5: Impact of varying prompt-based GNN layers on node classification performance.

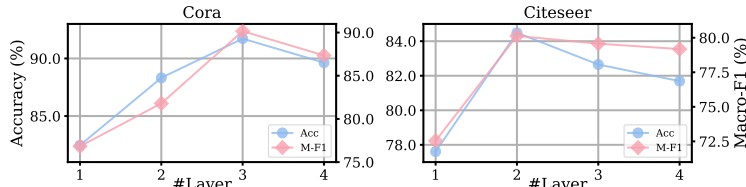

Figure 6: Ablation studies on node classification performance across three datasets.

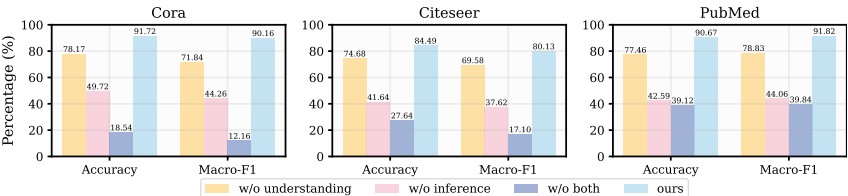

**Cross-task Generalization.** We also explore the transferability from link prediction to node classification. Table 4 summarizes the cross-task performance. As expected, our zero-shot variant LP→NC underperforms compared to supervised learning, but consistently exhibits significant improvement over the w/o training variant across all three datasets, indicating the adaptability to unseen tasks of PromptGFM. Overall, these experiments highlight that PromptGFM effectively transfers global knowledge across graphs and tasks, qualifying it as a versatile and knowledgeable GFM.

### 5.3 ABLATION AND EXPLORATION STUDIES

**Ablation Studies.** To investigate the contributions of each module, we design the following variants: **(1) w/o understanding.** This variant removes the prompt-based GNN and uses only the summarized textual representations of nodes for LLM fine-tuning. **(2) w/o inference.** This variant retains the GNN replication but performs LLM-based inference without additional fine-tuning. **(3) w/o both.** This variant relies solely on the initial textual summaries and uses pre-trained LLMs for inference without any GNN integration or fine-tuning. Figure 6 illustrates the node classification accuracy of these variants in Cora, Citeseer, and PubMed, with our full model consistently outperforming in all settings. First, the decrease in w/o understanding variant suggests that the absence of GNN replication results in the loss of crucial semantic and structural information. Second, the w/o inference variant exhibits a significant decline, highlighting the critical role of fine-tuning in integrating specific knowledge into the pre-trained model. Lastly, the w/o both variant yields the worst results, underscoring the synergy between the understanding and inference modules for overall performance.

**Hyperparameter Sensitivity.** We explore the impact of the number of layers in our prompt-based GNN. As shown in Figure 5, we can observe that PromptGFM progressively improves as the layer increases due to its ability to capture broader context and higher-order relationships over the graph. However, after a certain point, further stacking layers results in diminishing returns or even performance degradation due to over-smoothing, where node representations become indistinguishable within their local structures. This trend is consistent with traditional GNNs. Optimal performance is achieved with 3-layer GNN for Cora, while Citeseer reaches its best results with two layers. This analysis suggests that textual representations can be propagated over the graph similarly to numerical embeddings, effectively capturing semantic and structural information simultaneously.

**Case Study.** To provide further insights, we provide a case study showing the layer-by-layer refinement of textual representations in Appendix F.1. Additionally, since the nodes represent research papers, we compare their language-based IDs with key words from the papers, showing the superiority of PromptGFM in capturing essential meanings. Details can be found in Appendix F.2 .

## 6 CONCLUSION

We present PromptGFM, a graph foundation model grounded in graph vocabulary learning. By replicating the GNN workflow within the language space, we decouple the refined textual representations of nodes and establish an expressive and universal graph vocabulary. This vocabulary endows compatibility and scalability with natural language, enabling seamless transferability across graphs and tasks. Experiments demonstrate superior overall performance and strong cross-graph and cross-task generalization. Our research reveals the potential to function LLM as GNN and opens new avenues to build GFMs within the language space.

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

## SUMMARY OF THE APPENDIX

This appendix contains additional details for the ICLR 2025 submission, titled *"LLM as GNN: Graph Vocabulary Learning for Graph Foundation Model"*. The appendix is organized as follows:

- §A provides **Data Descriptions** used in our experiments.
- §B illustrates more details about **Baselines** employed for comparison.
- §C analyzes **Prompt Design** in our PromptGFM framework.
- §D shows more **Implementation Details**.
- §E reports the **Extended Experiments** on node classification.
- §F presents the **Case Study** to offer deeper insights of our research.
- §G gives **Announcement for LLM Selection** in our experiments for reference.

## A  DATA DESCRIPTIONS

Table 7: Statistics of four public benchmarking datasets for our research.

| Dataset | #Nodes | #Edges | #Labels | Domain |
|---------|--------|--------|---------|--------|
| Cora | 2,708 | 5,429 | 7 | Computer Science |
| Citeseer | 3327 | 4732 | 6 | Computer Science |
| Pubmed | 19,717 | 44,338 | 3 | Biomedical |
| ogbn-arxiv | 169,343 | 1,166,243 | 40 | Computer Science |

We utilize four public benchmarking datasets to evaluate our framework, including Cora, Citeseer, PubMed, and obgn-arxiv. The statistics of these datasets is illustrated in Table 7. We provide detail information as follows.

● **Cora** (McCallum et al., 2000). The dataset include a citation network consisting of 2,708 scientific publications in the field of machine learning, categorized into 7 classes based on their research topics. Nodes represent individual papers, and edges denote citation links between them, totaling 5,429 connections. Each paper is described the paper contents (titles and abstracts).

● **Citeseer** (Giles et al., 1998). This work introduces a citation network dataset comprising 3,327 scientific publications, categorized into 6 classes, including Agents, Artificial Intelligence, Database, Information Retrieval, Machine Learning, and Human-Computer Interaction. Nodes correspond to documents, and edges represent citation relationships between them, amounting to 4,732 links.

● **PubMed** (Sen et al., 2008). PubMed is a citation network of 19,717 scientific publications from the PubMed database pertaining to diabetes, classified into 3 classes: experimental induced diabetes, type 1 diabetes, and type 2 diabetes. Nodes are research papers, and edges signify citation links, amounting to 44,338 connections. This dataset is used for large-scale graph representation learning and evaluating algorithms in the biomedical domain.

● **obgn-arxiv** (Hu et al., 2020). The ogbn-arxiv dataset is part of the Open Graph Benchmark and consists of a directed citation graph of 169,343 arXiv papers, categorized into 40 subject areas. Nodes represent individual papers, and edges indicate citation relationships, totaling over 1.1 million connections. Each paper includes textual data from its title and abstract.

## B    BASELINES

We provide detailed information on the baseline models, categorized into: (1) Graph-agnostic methods, (2) GNN-based methods, (3) Transformer-based methods, and (4) GNN-LLM integration methods.

### B.1    GRAPH-AGNOSTIC METHODS.

• **MF.** This baseline is a matrix factorization approach with Bayesian personalized ranking as the objective function. It learn dense embeddings to reconstruct the adjacent matrix. For both node classification and link prediction, We adapt it to predict the labels of nodes or edges, respectively.

• **MLP.** This method adopts a multi-layer perceptron to learn low-dimensional embeddings for each node. In our work, we randomly initialize node embeddings without textual attributes for both graph-agnostic methos.

### B.2    GNN-BASED METHODS.

• **GCN** (Kipf & Welling, 2017). This model introduces a neural network architecture that generalizes convolution operations to graph-structured data, enabling effective semi-supervised learning by aggregating feature information from a node's local neighborhood.

• **GAT** (Velickovic et al., 2017). This method incorporates attention mechanisms into graph neural networks, allowing nodes to assign different importance weights to their neighbors during feature aggregation, which enhances performance by focusing on the most relevant connections.

• **SAGE** (Hamilton et al., 2017). GraphSAGE is an inductive representation learning framework on large graphs; it generates node embeddings by sampling and aggregating features from a node's local neighborhood, facilitating generalization to unseen nodes or graphs.

• **ReVGNN** (Li et al., 2021). This method includes a recurrent graph neural network tailored for dynamic graphs, capturing temporal dependencies by updating node representations as events occur over time, which is crucial for modeling evolving graph structures.

• **AGNN** (Thekumparampil et al., 2018). AGNN leverages attention mechanisms to compute attention coefficients based on node features, enabling the network to dynamically weigh the influence of neighboring nodes without introducing additional parameters.

• **DNA** (Fey, 2019). This work presents a flexible neighborhood aggregation method that dynamically selects and combines information from variable-sized node neighborhoods, enhancing the expressive power and adaptability of graph neural networks.

### B.3    TRANSFORMER-BASED METHODS.

• **SGFormer** (Wu et al., 2023). This work introduces a transformer-based architecture designed for graph data, integrating spectral graph theory into the transformer framework. It aims to capture both local and global graph structures efficiently by incorporating spectral filters, enhancing the model's ability to learn complex graph representations.

• **NodeFormer** (Wu et al., 2022). This framework presents a scalable graph transformer model that utilizes a randomized attention mechanism to approximate full attention on graphs. By reducing computational complexity, it enables efficient learning on large-scale graphs while preserving the expressiveness of transformer architectures.

### B.4    GNN-LLM INTEGRATION METHODS.

• **LLaGA** (Chen et al., 2024a). This model effectively integrates LLM capabilities to handle the complexities of graph-structured data. It transforms graph nodes into structure-aware sequences and maps them into token embedding space using a specialized projector. LLaGA excels in generalization and interpretability, performing strongly across various datasets and tasks. It also supports zero-shot learning, making it highly adaptable for unseen datasets.

- **OFA** (Liu et al., 2024a). This paper proposes a framework that handles various graph classification tasks across different domains using a single model. It introduces the nodes-of-interest (NOI) subgraph mechanism to standardize different tasks with a single task representation. Additionally, a novel graph prompting paradigm to leverage in-context learning and apply the same architecture across diverse graph classification tasks, achieving generalization across multiple domains.

- **GraphPrompter** (Liu et al., 2024b). This work introduces a novel framework designed to align graph with LLMs via soft prompts. Specifically, it adopts GNNs to capture graph structure and leverages an LLM to interpret the textual information at the node level. By prompt tuning, this approach demonstrates the potential of LLMs to effectively interpret graph structures, combining both semantic and structural insights for improved graph learning tasks.

- **ENGINE** (Zhu et al., 2024). This paper proposes a parameter- and memory-efficient fine-tuning method for textual graphs by using LLMs as encoders. It combines the LLMs and GNNs through a tunable GNN-based side structure, called G-Ladder, alongside each LLM layer, effectively reducing training costs without compromising performance.

## C  PROMPT DESIGN

In this section, we provide the templates of prompts in our PromptGFM framework.

---

**Prompt for node summarization.**

The title of the paper is <the title of the paper>, the
abstract of the paper is <the abstract of the paper>.  Please
summarize the paper.

---

**Prompt for each GNN layer replication.**

Given the central node <l-th round textual representation
of the central node>.  The selected one-hop neighbors are
[< l-th round of node #1>, <l-th round node #2>, ...  ,<l-th
round node #N>].  Please aggregate neighbor nodes and update
a concise yet meaningful representation for the central node.
Note connected nodes should share similar semantics and vice
versa.

---

**Prompt for node classification.**

<the language-based ID of the central node> has 1-hop
connections with [..., <language-based IDs of its 1-hop
neighbors>, ...], and it also has 2-hop connections with
[..., <language-based IDs of its 2-hop neighbors>, ...].
Which category should <the language-based ID of the central
node> be classified as ?

---

**Prompt for discriminative link prediction.**

<the language-based ID of the central node> has 1-hop
connections with [..., <language-based IDs of its 1-hop
neighbors>, ...], and it also has 2-hop connections with
[..., <language-based IDs of its 2-hop neighbors>, ...].
Among <the language-based ID of the central node> and <the
language-based ID of its negative sampling node>, which node
will be connected to <the language-based ID of the central
node>?

---

**Prompt for generative link prediction.**

<the language-based ID of the central node> has 1-hop
connections with [..., <language-based IDs of its 1-hop
neighbors>, ...], and it also has 2-hop connections with
[..., <language-based IDs of its 2-hop neighbors>, ...].
Which node should be connected to <the language-based ID of
the central node>?

## D  IMPLEMENTATION DETAILS

We provide further information for reproduction. In the graph understanding module, we selected the number of layers for the prompt-based GNN from $\{1, 2, 3, 4\}$. We randomly sampled 30% of the first-order neighbors during neighborhood sampling, capping the maximum number of sampled nodes at 12 to reduce computational cost and prevent overfitting. In the graph inference module, we fine-tuned the LLM with a learning rate of 3e-4 and a batch size of 4. To mitigate potential biases introduced by task-specific prompts, we designed a prompt pool for each task requirement and randomly selected prompts during instruction construction to enhance robustness. We employed a standard early-stopping strategy during training: if the performance metric on the validation set did not improve over a fixed number of consecutive epochs (determined based on the dataset), we halted training to prevent overfitting. For other hyperparameters of the compared methods, we referred to the original papers and carefully tuned them to suit each dataset.

## E  EXTENDED EXPERIMENTS

Due to space constraints in the main text, we provide the comparison of node classification Macro-F1 scores across four categories of baselines in Table 8. Unfortunately, OFA, LLaGA, ENGINE, and GraphPrompter did not report F1-scores in their respective papers.

Table 8: Comparison with four categories of baselines on node classification Macro-F1 scores. The metrics for GNN-LLM models are unavailable in their respective papers.

| Model | Cora | Citeseer | PubMed | ogbn-arxiv |
|---|---|---|---|---|
| MF | 58.26±2.81 | 55.17±1.97 | 73.15±0.79 | 39.81±0.58 |
| MLP | 59.44±2.96 | 57.85±2.26 | 75.23±0.91 | 41.79±0.32 |
| GCN | 79.20±5.80 | 70.30±2.68 | 79.95±1.86 | 39.94±1.24 |
| GAT | 80.12±3.60 | 67.00±1.71 | 79.40±2.04 | 35.35±0.68 |
| SAGE | 81.96±3.45 | 66.73±4.03 | 81.58±1.08 | 46.81±2.13 |
| RevGNN | 85.24±2.85 | 71.12±2.00 | 83.94±1.29 | 46.18±0.74 |
| AGNN | 75.71±2.70 | 64.03±2.36 | 75.69±1.77 | 29.48±3.35 |
| DNA | 76.63±5.77 | 63.22±2.27 | 80.86±1.23 | 20.94±0.64 |
| SGFormer | 79.28±4.85 | 63.42±2.37 | 79.31±1.72 | 44.89±3.79 |
| NodeFormer | 77.98±2.91 | 62.58±2.19 | 77.35±1.35 | 43.87±1.05 |
| PromptGFM | **91.72±1.06** | **84.49±1.37** | 90.67±1.16 | **80.58±0.54** |

## F  CASE STUDY

### F.1  TEXTUAL REPRESENTATIONS IN PROMPT-BASED GNNS

In this part, we select two representative cases in citation networks and demonstrate their layer-by-layer refinement of our prompt-based GNN. Specifically, we provide the textual representations at each layer, including Round 0 as initial features. More importantly, we also collect its one-hop nodes and annotate the source and relevant information below (highlighted in blue). From our empirical studies, we have the following observations. Overall, it is evident that the verbose textual representations are progressively refined to concise textual presentations. Meanwhile, the core semantics become increasingly clear throughout the process, until a short sequence composed of several natural language tokens at the last round. Furthermore, we notice that we effectively incorporate the key ideas of some neighboring nodes, as reflected in the refined textual representations after each aggregation-update operation. More specifically, the neighboring information is naturally fused to the central node within the language space, boosting their connections and improving the expressiveness of the central node. This study further illustrates the explicit message passing through our proposed prompt-based GNN in the language space, which captures the graph semantics and structures simultaneously.

**Case A.**

**Raw text attributes:** Paper Title: Evaluating Neural Network Predictors by Bootstrapping. Abstract: We present a new method, inspired by the bootstrap, whose goal it is to determine the quality and reliability of a neural network predictor. Our method leads to more robust forecasting along with a large amount of statistical information on forecast performance that we exploit. We exhibit the method in the context of multi-variate time series prediction on financial data from the New York Stock Exchange. It turns out that the variation due to different resamplings (i.e., splits between training, cross-validation, and test sets) is significantly larger than the variation due to different network conditions (such as architecture and initial weights). Furthermore, this method allows us to forecast a probability distribution, as opposed to the traditional case of just a single value at each time step. We demonstrate this on a strictly held-out test set that includes the 1987 stock market crash. We also compare the performance of the class of neural networks to identically bootstrapped linear models.

**Round 0:** This method evaluates neural network predictors using a bootstrap-inspired approach, improving robustness in NYSE financial data forecasting. It predicts probability distributions rather than single values, surpassing resampling variations. Compared to bootstrapped linear models, it enhances accuracy, as demonstrated during the 1987 stock market crash.

**Round 1:** This bootstrap method improves neural network evaluations in NYSE forecasting, **comparable to traditional error estimation methods. It clears and simplifies data structures**, enhancing accuracy and predicting probability distribution.
/* **"comparable to traditional error estimation methods"** comes from paper: *A comparison of some error estimates for neural network models*; **"It clears and simplifies data structures"** comes from paper: *The Observer-Observation Dilemma in Neuro-Forecasting: Reliable Models From Unreliable Data Through CLEARNING..* */

**Round 2:** This bootstrap-inspired method refines neural network evaluations for NYSE forecasting, highlighting **local error bars from nonlinear regression analysis**, improving accuracy with data simplification techniques.
/* **"local error bars from nonlinear regression analysis"** comes from paper: *Learning Local Error Bars for Nonlinear Regression*/

**Round 3:** Bootstrap method refines neural forecasts, integrates error bars, enhances accuracy.

**Case B.**

**Raw text attributes:** Paper Title: Insulin dependent diabetes in Asians. Abstract: Type 1 diabetes is said to be extremely rare in children in India, where diabetes treated with insulin may be due to chronic pancreatic disease or malnutrition. To see whether typical type 1 diabetes occurred in Asian children in the United Kingdom, all known Asian children with diabetes in industrial West Yorkshire were ascertained. A total of 17 such children were studied; of these, seven were from three multiplex families and two fathers from these families had diabetes. All children were ketosis prone and developed diabetes while resident in the UK. There were significant increases in HLA-B8 and HLA-DR3 and increases in HLA-DR4 and HLA-DR3/DR4, while HLA-B15 was absent. Islet cell antibodies, either IgG or complement fixing, were present in four of 18 subjects tested, all of whom had disease of short duration. The prevalence of type 1 diabetes in Asian children aged 15 years or less in West Yorkshire was 36/100,000, assuming complete ascertainment. It is concluded that typical type 1 diabetes may occur in Asian children and this condition may be more common in families who have migrated to the UK.

**Round 0:** The study examined type 1 diabetes prevalence in 17 Asian children in West Yorkshire, UK, focusing on family history. Seven were from multiplex families, and all were ketosis-prone. Significant increases in HLA-B8, HLA-DR3, and HLA-DR4 were found, with no HLA-B15. Islet cell antibodies were present in four of 18 subjects. The diabetes prevalence was 36/100,000, suggesting it may be more common in migrant families.

**Round 1:** The study examined type 1 diabetes in Asian children in West Yorkshire, UK. **Environmental factors and migration may raise incidence**, unlike the **low incidence of insulin-dependent diabetes in Karachi**. Increases in HLA-B8, HLA-DR3, and HLA-DR4 were found. /*__"Environmental factors and migration may raise incidence"__ comes from paper: *__Evidence for an environmental effect in the aetiology of insulin dependent diabetes in a transmigratory population__*; __"low incidence of insulin-dependent diabetes in Karachi"__ comes from paper: *__Incidence of insulin dependent diabetes mellitus in Karachi, Pakistan__*/

**Round 2:** The study examined type 1 diabetes in Asian children in West Yorkshire, UK. **Migration may raise incidence**, unlike the low incidence in Karachi. Increases in HLA markers were found, similar to **North and South Indian diabetics**. /*__"Migration may raise incidence"__ comes from the paper: *__HLA-DR antigen frequencies in a North Indian type I diabetic population__*; __"North and South Indian diabetics"__ comes from papers: *__HLA-DR antigen frequencies in a North Indian type I diabetic population__* and *__HLA, complement C2, C4, properdin factor B and glyoxalase types in South Indian diabetics__*. */

**Round 3:** Type 1, diabetes. Migration and HLA markers linked to increased diabetes incidence.

## F.2 LANGUAGE-BASED IDs VS. KEY WORDS

To provide further insights, we leverage external information to validate the superiority of PromptGFM in capturing the core semantics of nodes. Since all datasets in this work represent research publication network, we extract key words from the papers and compare them with their language-based IDs in our universal graph vocabulary. Table 9 summarizes the key words and language-based IDs of selected papers, along with their titles and URLs for reference. Overall, it is evident that there are strong semantic relevance between the language-based IDs and keywords. For example, regarding the paper titled *Distributed Protocols at the Rescue for Trustworthy Online Voting*, the key words have appeared within its language-based ID, suggesting that PromptGFM has effectively captured its core semantics through our prompt-based GNN. In addition, in *Committees providing EJR can be computed efficiently*, where the title is less indicative of the content, the language-based ID still aligns perfectly with the corresponding key words, such as *efficient computation* and *rules*. This finding demonstrates that PromptGFM not only effectively captures the core idea without relying on the title, but also filters relevant semantics from neighboring nodes to enhance its own representations. Overall, our language-based IDs accurately capture and extend the semantics of the nodes, making them well-suited to form a universal graph vocabulary.

## G ANNOUNCEMENT FOR LLM SELECTION

Furthermore, we acknowledge the rapid advancements in LLMs. In the graph understanding module, employing more powerful models like GPT-4o and GPT-o1 could enhance the reproduction of the GNN flow and generate higher-quality textual representations. Similarly, in the graph inference module, fine-tuning larger open-source LLMs, such as LLaMA, may lead to improved results due to their increased capacity to model complex patterns. While integrating these advanced models holds promise for better performance, it also introduces additional computational requirements and challenges in fine-tuning. We leave the exploration of these possibilities as future work. Conversely, achieving state-of-the-art performance using previous models further highlights the robust design of our graph foundation model.

Table 9: The comparison between the language-based IDs from the graph vocabulary and the key words in their original paper. The observed similarity demonstrates that our prompt-based GNN effectively captures the essential meanings of these nodes.

| Paper | Language-based ID | Key Words | URL |
|---|---|---|---|
| Modular Verification of Interrupt-Driven Software | Modular verification of interrupt-driven software using abstract interpretation | Software, Abstract Interpretation, Feasibility Verification | arXiv:1709.10078 |
| Parsimonious Data: How a single Facebook like predicts voting behaviour in multiparty systems | Predicting voting behavior using Facebook likes in multiparty systems | Facebook Likes, Voter Intention, Machine Learning, Multiparty System | arXiv:1704.01143 |
| A Fast Noniterative Algorithm for Compressive Sensing Using Binary Measurement Matrices | Fast noniterative algorithm for compressive sensing with binary matrices | Compressive Sensing, Deterministic Methods | arXiv:1708.03608 |
| Optimization of Battery Energy Storage to Improve Power System Oscillation Damping | Battery storage optimization improves power system oscillation damping | Battery Energy Storage System, Oscillation Damping | arXiv:1811.10213 |
| Neural Variational Hybrid Collaborative Filtering | Neural Variational Hybrid Collaborative Filtering improves recommendation performance | Collaborative Filtering, VAE, Recommendation System | arXiv:1810.05376v6 |
| Interpretable Neural Networks for Predicting Mortality Risk using Multi-modal Electronic Health Records | Predicting mortality risk using interpretable multi-modal neural network | Mortality Risk Prediction, Clinical Data | arXiv:1901.08125 |
| A New Approach to Distributed Hypothesis Testing and Non-Bayesian Learning: Improved Learning Rate and Byzantine-Resilience | Distributed hypothesis testing with Byzantine resilience using Bayesian update | Bayesian Learning, Byzantine Resilience | arXiv:1907.03588 |
| Accurate and Efficient Hyperbolic Tangent Activation Function on FPGA using the DCT Interpolation Filter | Efficient hyperbolic tangent activation function using DCTIF | Hyperbolic Tangent, Activation Function | arXiv:1609.07750 |
| Distributed Protocols at the Rescue for Trustworthy Online Voting | Trustworthy online voting with distributed blockchain protocols | Distributed Voting, Distributed Protocols | arXiv:1705.04480 |
| Committees providing EJR can be computed efficiently | Efficient computation of approval-based multi-winner voting rules | Approval-Based Voting, Multi-Winner Elections | arXiv:1704.00356 |
| Creatism: A deep-learning photographer capable of creating professional work | Creatism: deep learning system for artistic photography creation | Creatism, Evaluation of Photographic Quality, Deep Learning | arXiv:1707.03491 |
| Relation of familial patterns of coronary heart disease, stroke, and diabetes to subclinical atherosclerosis: the multi-ethnic study of atherosclerosis | Family history beyond early-onset heart disease impacts atherosclerosis | Family History, Coronary Heart Disease, Stroke | doi.org/10.1097/GIM.0b0 13e31818e639b |
| Glycemic index, glycemic load, and risk of type 2 diabetes | Benefits of low-GI diet in type 2 diabetes | Diabetes, Prevention | doi.org/10.1093/ajcn/76/1. 274S |
| Decreased insulin responsiveness of glucose uptake in cultured human skeletal muscle cells from insulin-resistant nondiabetic relatives of type 2 diabetic families | Inherited defects contribute to insulin resistance in diabetes | Insulin Resistance, Inherited Factors | doi.org/10.2337/diabetes. 49.7.1169 |
| Quantitative histopathological studies of the extramural coronary arteries from Type 2 (non-insulin-dependent) diabetic patients | Histopathological study of coronary arteries in diabetic patients | Histopathology, Diabetes Mellitus | doi.org/10.1007/BF0027 4798 |
| Metabolic control and diet in Finnish diabetic adolescents | Factors influencing metabolic control in diabetic adolescents | Diabetes Mellitus, Adolescent | doi.org/10.1111/j.1651-2227.1992.tb12212.x |

