# OpenReview forum: "LLM as GNN: Graph Vocabulary Learning for Graph Foundation Model"
_ICLR.cc/2025/Conference — ICLR 2025 Conference Withdrawn Submission_

### Official Review · Reviewer_gQ3P · 2024-10-27

**Soundness:** 2
**Presentation:** 2
**Contribution:** 2
**Rating:** 3
**Confidence:** 5

**Summary:**

The paper presents PromptGFM, an approach for integrating LLMs with GNNs by developing a language-based graph vocabulary. It aims to resolve limitations in current GNN-LLM architectures and demonstrates competitive performance in node classification and link prediction.

**Strengths:**

1. Novel attempt at LLM-GNN unification using natural language as a graph vocabulary.
2. Promising results on basic benchmarks.

**Weaknesses:**

1. The paper claims applicability to all graph types, though it only demonstrates effectiveness on text-attributed graphs.
2. Lacks evidence that the method generalizes to non-textual graphs, which is critical given the claim of a "universal" graph model.
3. How does the model handle graphs without inherent text attributes?
4. Can the authors provide clarity on novel contributions beyond combining existing techniques?

**Questions:**

See weaknesses.

---

> ### Author Response · Authors · 2024-11-24
> **Response to Reviewer gQ3P**
>
> Thanks for your review. We clarify your misunderstanding as follows.
>
> W1. Apologies for the confusion. As discussed in the Introduction, PromptGFM is specifically designed for text-attributed graphs and can be applied to any type of text-attributed graph, regardless of the domain.
>
> W2. Sorry for confusion. We would like to clarify our work aims to do graph foundation model for text-attributed graphs only. After multi-round message passing, each node can be represented by a finite sequence of language tokens and has embodied structural information, which is called language-based IDs. As language tokens are universal and transferable in NLP, the language-based IDs for nodes are also universal across different text-attributed graphs. With universal textual representations, we can construct a large number of task-specific instructions using pure language tokens, and put them in a single multi-instruction fine-tuning framework, adapting to various graphs and tasks. This is how our method works as a universal graph model.
>
> W3. Sorry for your misunderstanding. As mentioned in the Introduction Section, our proposed PromptGFM is a graph foundation model for text-attributed graphs, which is the most studied setting in the domain. We will highlight this setting for better clarification.
>
> W4. Thanks for your question. We highlight our contributions as follows. (1) Our work represents an attempt at LLM as GNN, where we use prompts to guide LLMs in replicating message passing for each node, aligning with the fundamental principles of existing GNN models. This approach demonstrates how LLMs can function as GNNs, fundamentally differing from existing LLM for GNN and GNN for LLM models, where GNNs and LLMs operate independently. Our approach represents an entirely new paradigm and an unexplored concept; it is by no means a simple direct combination of LLM and GNN. (2) By implementing LLM as GNN in the textual space, our approach addresses out-of-vocabulary issues present in previous works and enables the use of pure language prompt instructions for LLM inference. This resolves modality incompatibility and facilitates positive transfer, significantly advancing the development of GFMs.

---

> > ### Comment · Reviewer_gQ3P · 2024-11-25
> >
> > Thanks for your reply. But you just call your model a "graph foundation model" in your title. I suggest maybe just call it "graph-llm".

---

### Official Review · Reviewer_xm9X · 2024-10-27

**Soundness:** 1
**Presentation:** 3
**Contribution:** 1
**Rating:** 3
**Confidence:** 4

**Summary:**

The paper introduces PromptGFM, an LLM-based approach to perform node classification and link prediction on text-attributed graphs (TAGs) only. First, PromptGFM uses one LLM to summarize textual node features (like a title and abstract of a paper) into a “Language-based ID” string. Second, another LLM is fine-tuned on prompts with this textual node ID and a sub-sample of IDs of its one-hop neighbors to “perform neighbor aggregation” and predict a label for node classification or node ID for link prediction.

**Strengths:**

S1. Intra- and inter-domain transferability experiments might be of certain interest.

**Weaknesses:**

**W1.** The paper completely lacks theoretical and experimental support for its elaborated claims such as “meticulously design a series of prompts to align with the GNN workflow at the finest granularity”, “faithfully reproduce the message passing paradigm of GNNs”, “concise yet meaningful representations”, “we capture semantic and structural information through the prompt-based GNN”, “generate expressive representations”.

There is no formal study, proofs, or experimental analysis of how LLM prompts like “please aggregate the following neighbors” can ever capture the math of message passing or its results. Or how “meaningful” or “expressive” the LLM representations are compared to GNNs (there is a whole body of literature on the theory of GNN expressiveness that might have been of help). Perhaps the biggest mistake in claiming the alignment with GNNs is the fact that GNNs are permutation-invariant models whereas all autoregressive LLMs are by default *permutation-variant*, that is, the result of “Please aggregate <node 1> and <node 2>” is very likely be different from “Please aggregate <node 2> and <node 1>” (at least due to positional encodings which will featurize the nodes differently). Constructing a few prompts and claiming “faithful reproduction” without any formal study or theoretical guarantees is not enough to sell such claims.

Similarly, claiming in the ablations (Section 5.3) that PromptGFM suffers from over-smoothing after 4 “message-passing layer” prompting steps has no theoretical or experimental evidence - since there is no notion of a discrete node in PromptGFM (a node is represented with several tokens from the LLM vocab), then what is getting oversmoothed? There are rather rigorous mathematical analyses of oversmoothing [1,2] that measure the distance between node representations of different layers - it would require evidence of a similar phenomenon in scattered tokenized LLM representations to claim oversmoothing in this case.

[1] Oono, Suzuki. Graph Neural Networks Exponentially Lose Expressive Power for Node Classification. ICLR 2020
[2] Southern, Di Giovanni, et al. Understanding Virtual Nodes: Oversmoothing, Oversquashing, and Node Heterogeneity

**W2.** The paper largely oversells its technical contributions, namely, the Graph Understanding Module that replicates message passing (see **W1** why it does not, there is no evidence of such replication); and the universal graph vocabulary which works “across all the graphs and tasks” - in fact, PromptGFM does not propose any new vocabulary for encoding graphs and just relies on the existing LLM token vocabularies and textual descriptions of nodes. If input graphs do not have textual features (a common case for non-citation graphs), PromptGFM appears to be of questionable value as a graph foundation model. The paper repeatedly claims that “existing methods treat nodes as OOV tokens” whereas the vast majority of “LLMs for graphs” approaches (including compared OFA or GraphText) do exactly the same as PromptGFM and use textual node features as part of an LLM prompt.

**W3**. The experimental agenda is rather underwhelming and raises a lot of questions about the practical applicability of PromptGFM.

* Only 4 standard citation datasets for node classification (NC) and link prediction (LP);
* Link prediction experiments employ GNN baselines unsuited for this task - the authors are aware of the benchmark by Chen et al, 2024 which consists of 20 node/link/graph-level tasks and used much stronger baselines like BUDDY for link prediction;
* Comparing billion-sized components of PromptGFM (GPT 3.5 Turbo for Language Node IDs + fine-tuned T5 for actual tasks) which need several GPUs and high monetary inference costs even for small standard graph datasets like Cora vs much smaller GNNs (often 100-1000x smaller) that run for free even on CPUs presents quite a myopic and biased perspective on the advantages of LLMs for graph learning tasks;
* It is hard to quantify the importance of reported results when many important experimental details are missing. Are node labels in the NC task encoded via text or somehow else? Are LLMs asked to predict a text label of a node label or select one of K options? How many negative samples were used in the LP task? What is the size of T5 fine-tuned on the NC and LP tasks (there are many options)? Was it a full fine-tune or LoRA?

**Questions:**

- Are node labels in the NC task encoded via text or somehow else?
- Are LLMs asked to predict a text label of a node label or select one of K options?
- How many negative samples were used in the LP task for PromptGFM and GNN baselines?
- What is the size of T5 fine-tuned on the NC and LP tasks (there are many options)?
- Was it a full fine-tune or LoRA?

---

> ### Author Response · Authors · 2024-12-04
> **Response to Reviewer xm9X**
>
> Thank you so much for your review. In the following, we clarify some of your misunderstanding.
>
> W1. Thank you for your thoughtful feedback. Our work introduces a novel approach to implementing GNN functionality within the natural language space using LLMs. This approach is not intended to entirely replace embedding-based GNNs but rather to offer a complementary perspective that explores new possibilities for leveraging LLMs in graph-related tasks. Consistent with our motivations, we argue that in the era of LLMs, providing formal mathematical proofs may not always be strictly necessary. Instead, our focus on practical and conceptual alignment has already demonstrated meaningful and promising results. We will try to clarify our points to avoid misunderstanding.
>
> W2. We would like to clarify that our work focuses specifically on building a graph foundation model for text-attributed graphs. In our approach, each node is represented by a textual description, referred to as a language-based ID, which ensures applicability and comparability across all text-attributed graphs. Unlike existing methods, we highlight that our graph vocabulary, composed of natural language tokens, is inherently sharable and transferable across different graphs, whereas approaches using out-of-vocabulary (OOV) tokens limit generalization to specific graph nodes. Furthermore, our work is fundamentally different from most existing methods and represents the first true implementation of "LLM as GNN," as outlined in the introduction. For instance, OFA defines functional nodes and uses an LLM to convert textual descriptions into embeddings but does not utilize the LLM for core GNN operations such as message passing. Instead, in OFA, message passing is performed in the vector space using traditional GNN models, leading to embeddings that lack interpretability and generalizability to other graphs or tasks. Our approach directly addresses these limitations by leveraging LLMs for the entire GNN workflow, enabling both interpretability and broader applicability.
>
> W3. Our experimental design aligns with standard practices in the field, as demonstrated by studies such as [1][2][3], which follow widely accepted benchmark setups. Regarding resource consumption, we acknowledge that our model requires more computational resources compared to traditional GNNs. However, it delivers higher accuracy and focuses on foundation models with transferability and generalization capabilities, enabling cross-dataset and cross-task applications—advantages that traditional GNNs cannot achieve.
> Regarding the experimental details, please check our responses to the questions Q1-Q5.
>
> Q1. In the graph understanding module, we only prompt the LLM to perform message passing without using labels. In fact, the performance of different downstream graph tasks depends on different aspects of the graph (node classification tasks focus on the smoothness of nodes, while link prediction tasks emphasize local structural information). Therefore, this approach ensures the outputs of the graph understanding module can generalize across both tasks, i.e. node classification and link prediction.  In the graph inference module, when we need construct task-specific prompts, node labels are directly provided in the prompts for node classification tasks to avoid hallucination.
>
> Q2. For node classification, we directly provide the candidate labels and ensure that LLMs can be select one of them. This design is aligned with traditional node classification settings.
>
> Q3. As mentioned in Section 4.1, we rely on prompts "Note connected nodes should share similar semantics and vice versa" to intuitively optimize the textual representations because negative sampling is redundant in the situation. As for GNN baselines, the number of negative samples is set to 1 by default.
>
> Q4. The size of the T5 model fine-tuned on the NC and LP tasks is 0.8B.
>
> Q5. It was a full fine-tune. Since Flan-T5-large has relatively few parameters, we perform full-parameter fine-tuning.

---

### Official Review · Reviewer_9vAr · 2024-10-28

**Soundness:** 2
**Presentation:** 2
**Contribution:** 2
**Rating:** 3
**Confidence:** 4

**Summary:**

This work proposes a graph foundation model, PromptGFM. Specifically, it includes a graph understanding module where the LLM is prompted to perform 'message passing,' similar to that in GNNs. Additionally, there is a Graph Inference Module, in which each node in the graph is mapped to text tokens, ensuring expressiveness, transferability, and scalability

**Strengths:**

It is a good idea to prompt the LLM to simulate message passing in a GNN. The design of feature transformation, message passing, and message aggregation all sound reasonable.

**Weaknesses:**

- I am not convinced that this qualifies as a graph foundation model. It is an interesting exploration of how to integrate LLMs and GNNs. However, from a methodological perspective, since PromptGFM prompts the LLM to simulate message passing, it requires text-only attributes and is unable to handle non-textual graphs. In the experiments, the inter-domain ability was demonstrated by transferring from Cora/Citeseer/Arxiv to PubMed. However, this is not a typical inter-domain setting, as they are all citation graphs. There might be shared patterns within citation graphs that contribute to the observed 'inter-domain' ability. I would need more experiments to be convinced of the 'inter-domain' ability.
- Lack of strong baseline models. Table 1 didn't include those strong baseline modesl, such as TAPE[1].

[1] He, Xiaoxin, Xavier Bresson, Thomas Laurent, Adam Perold, Yann LeCun, and Bryan Hooi. "Harnessing explanations: Llm-to-lm interpreter for enhanced text-attributed graph representation learning." arXiv preprint arXiv:2305.19523 (2023).

**Questions:**

- How the node is mapping into the LLM's vocabulary (eq5)? How many tokens are need for each node?

---

> ### Author Response · Authors · 2024-11-24
> **Response to Reviewer 9vAr**
>
> Thank you so much for your review.  In the following, we clarify the misunderstanding and address your concerns.
>
> W1. (1) Sorry for your misunderstanding. As mentioned in the Introduction Section, our proposed PromptGFM is a graph foundation model for text-attributed graphs, which is the most studied setting in this research field. (2) Regarding dataset domains, most works in this field focus on how different domains incorporate varying semantics rather than the specifics of dataset construction. For instance, a recent comprehensive survey [1] adopts a similar approach, categorizing Cora/Citeseer/Arxiv as the CS citation domain and assigning PubMed to the MISC domain. We will consider including more datasets in future work.
>
> [1] Chen, Zhikai, Haitao Mao, Jingzhe Liu, Yu Song, Bingheng Li, Wei Jin, Bahare Fatemi et al. "Text-space Graph Foundation Models: Comprehensive Benchmarks and New Insights." arXiv preprint arXiv:2406.10727 (2024).
>
> W2. Thank you for your suggestion. The table presents node classification performance, showing comparable results. 'N/A' indicates that the results are not reported in the original paper. Please note that TAPE cannot handle link prediction tasks.
>
> | Model | Cora | Citeseer | PubMed | obgn-arxiv |
> | -------- | -------- | -------- | -------- | -------- |
> | TAPE     | 92.90±3.07     | N/A     | 96.18±0.53     | 77.50±0.12     |
>
> Q1. Thanks for your question. (1) Following a comprehensive analysis in Section 4.2, we directly use the final textual representations as language-based IDs, which is composed of a finite sequence of langugae tokens. (2) We do not specify a fixed number of tokens for the final textual representation. However, in practice, it typically ranges from 10 to 20 tokens, depending on the number of layers of the prompt-based GNN.

---

> > ### Comment · Reviewer_9vAr · 2024-11-25
> > **Response to authros**
> >
> > Thank you for addressing my questions.
> >
> > I agree that the contribution of this work should focus on the graph foundation model for TAG rather than general graph foundation models.
> >
> > After incorporating TAPE as a baseline, it seems that PromptGFM only outperforms on 1 out of 4 datasets.
> >
> > Overall, I will maintain my current score, considering the overall contribution and model performance.

---

### Official Review · Reviewer_p4FX · 2024-11-03

**Soundness:** 2
**Presentation:** 3
**Contribution:** 2
**Rating:** 5
**Confidence:** 5

**Summary:**

This paper introduces a graph foundational model. The proposed method employs one LLM to replicate a GNN’s workflow through prompting and another LLM for fitting downstream tasks. Specifically, they replicate the message-passing and local aggregation processes of a multilayer GNN by summarizing input using an LLM, prompting sampled one-hop neighborhoods of nodes to the LLM, and prompting it for aggregation across the neighborhoods. To mitigate the problem of out-of-vocabulary (OOV) faced by LLMs observing unseen nodes, they introduce graph vocabulary learning by making language-based IDs. This graph vocabulary is used for making prompts for the inferencing LLM. Finally, to increase generalization the LLM of inference module is fine-tuned on different tasks and datasets by multi-instruction learning

**Strengths:**

- The paper introduces a novel approach by employing multilayer LLMs to replicate the message-passing process of GNNs.
- The authors tackle an important issue of OOV tokens in LLMs when applied to graph tasks.
- Experimental results demonstrate the superiority of their model compared to existing LLM + GNN architectures.
- The paper does a comprehensive review of the current methods and the baselines are state-of-the-art.

**Weaknesses:**

While the proposed method demonstrates clear superiority over current state-of-the-art techniques, there are several significant concerns regarding the paper that need to be addressed:

- While the use of multiple layers of LLMs to replicate message passing across multi-hop structures is a novel approach, the fundamental concept of prompting LLMs for message passing in 1-hop neighborhoods is well-explored with similar methods [1, 3]. The authors should clearly distinguish their “graph understanding” module from existing techniques. Notably, their method incorporates nodes sequentially, which raises practical concerns for large graphs due to the maximum input length limitations of LLMs and the associated performance decline when handling long sequences. For example, showing how long-range dependencies and long sequences can be captured by their understanding module would discriminate from previous works. Additionally, the authors do not provide a clear example of a template showing how a node's textual representation is constructed. The examples in Figure 3 and the Appendix are not sufficiently informative and differ significantly from the case study provided. A concrete example that aligns with the templates outlined in the paper would greatly enhance understanding of the method. Concerning this, the authors would provide special tokens, words, phrases or more details of matching input text to a unified template.
- The prompt examples provided in the paper, along with the case study, illustrate that the graph understanding module summarizes the input text sequentially across multiple rounds. However, in GNNs, information is propagated through message passing rather than summarized. As a result, the LLM has not effectively replicated the message-passing and aggregation processes. Additionally, because the graph understanding module utilizes a non-deterministic LLM, some nodes and their associated feature and structural information may be lost across multiple layers. Consequently, retrieving the embeddings of all nodes after several rounds of prompting becomes challenging. The paper does not address how this information preservation is ensured, especially since the output of the n-th layer of the LLM is expected to represent all input nodes. For example, to address information loss due to the non-deterministic nature of LLMs authors would keep records of node representations after each round of prompting LLM and do an overall aggregation.
- The generalization of the LLM module for inference might be limited to the tasks and datasets used for fine-tuning which is far from a universal vocabulary as claimed in the paper. Also, this type of multi-task learning is also studied with GNN modules trained on different tasks and datasets as proposed in [1, 2, 3]. Authors would provide evidence or arguments supporting their claim of a "universal vocabulary", particularly in comparison to existing multi-task learning approaches with GNNs.
- The term "prompt-based GNNs" can be misleading, as the underlying model is actually an LLM, not a GNN, and there are fundamental differences between GNN-based models and LLMs. This confusion is further compounded by the visualization in Figure 2, which portrays the current work as a combination of a GNN and an LLM, despite the absence of a GNN module in the model. To enhance clarity, it would be beneficial to revise the terminology and the visualization to better reflect the model's true nature. For example, they can call their method a "prompt-based information propagation" and also remove the "GNN" block from the figure and keep the single LLM.
- In the "Data Description" section, the paper states that the Cora, Citeseer, and PubMed datasets are "introduced" by this work. This wording is misleading, as these datasets are not contributions of the paper. Authors would instead use alternatives like "used", "utilized", "evaluated/experiments on", etc.
- The explanations of some components in the proposed method, particularly in the sections on "Graph Vocabulary Learning" and "GNN Replication with LLMs," are overly detailed, which can detract from the paper's fluency. The authors should consider summarizing these sections to better highlight the main contributions. Also, the detailed explanations can be moved to an appendix or supplementary material.


[1] Hao Liu, Jiarui Feng, Lecheng Kong, Ningyue Liang, Dacheng Tao, Yixin Chen, & Muhan Zhang (2024). One For All: Towards Training One Graph Model For All Classification Tasks. In The Twelfth International Conference on Learning Representations.

[2] Sun, X., Cheng, H., Li, J., Liu, B., & Guan, J. (2023). All in One: Multi-Task Prompting for Graph Neural Networks. In Proceedings of the 29th ACM SIGKDD Conference on Knowledge Discovery and Data Mining (pp. 2120–2131). Association for Computing Machinery.

[3] Bahare Fatemi, Jonathan Halcrow, & Bryan Perozzi (2024). Talk like a Graph: Encoding Graphs for Large Language Models. In The Twelfth International Conference on Learning Representations.

**Questions:**

- The discussion on generating graph vocabulary remains incomplete. Specifically, which module is responsible for creating the graph vocabulary: the graph understanding module or the graph inference module? Based on Figure 4 and the data flow depicted, it appears that the graph vocabulary is generated before the predictor LLM. However, the paper discusses this in Section 4.2, which focuses on the graph inference module. Could it be that the graph vocabulary is constructed through another process between the two modules?
- The distinction between a regular node textual feature and a language-based ID is unclear. In the case study, the language-based ID seems to be a rephrasing of the input textual features. How, do these language-based IDs address the OOV problem if they only replicate a rephrased version of input?
- The representation of node features in graphs like Cora and PubMed as text is not addressed. Given that these graphs contain high-dimensional features, creating an optimal textual representation from them is challenging. How are these features conveyed in the text domain?

---

> ### Author Response · Authors · 2024-11-24
> **Response to  Reviewer p4FX (1)**
>
> We truly appreciate your constructive feedback and suggestions. In the following, we clarify the misunderstanding and highlight our contributions.
>
> W1. Thanks for you comments. (1) Our method differs significantly from [1, 3] in several key aspects. Specifically, in OFA [1], the authors use an LLM to transform each node into a fixed-length vector and perform message passing over a prompted subgraph in the embedding space. In contrast, PromptGFM introduces two key innovations: (a) Message passing is conducted in the textual space instead of the embedding space, an approach unexplored in prior work. (b) By stacking multi-layer LLM prompts, each node can theoretically capture high-order signals with rich semantics across the entire graph, rather than being limited to a subgraph as in OFA. Regarding "Talk Like a Graph" [3], while it establishes a benchmark by encoding graph data as text, it focuses solely on reasoning tasks and neglects graph understanding, such as message passing. Additionally, this work directly inputs all graph structural information (e.g., adjacency matrix, social networks) into a single prompt. This approach is clearly constrained by the LLM's allowable input length, rendering it impractical for real-world applications. In contrast, PromptGFM explicitly conveys neighboring information node by node via prompts, effectively replicating message passing—an approach not explored in prior works. (2) In each prompt, we include only the textual representations of the central node and its neighboring nodes from the previous layer to generate the textual representations for the current layer. By stacking multiple rounds of prompts to LLMs, we effectively capture long-range dependencies across the graph, similar to the workflow of GNNs. Additionally, inspired by GraphSAGE, we sample a limited number of 1-hop neighboring nodes for each node's prompt. This approach stays within ChatGPT's length constraints and avoids practical issues associated with long sequences. (3) We have included the templates of prompts for different functionalities in Appendix C. By feeding the title and abstract of each paper, we prompt LLM to construct the initial textual representation of each node. The second prompt in Appendix C illustrate how we replicate the message passing of each GNN layer, which is exactly the same as Figure 3. Additionally, we decompose this prompt to show how it align with the authentic GNN in Figure 2. For more clear demonstration, we will add a case study in Appendix F.1 to make it align with the example in the main content. It shows how we perform layer-by-layer message passing in the textual space and how information is aggregated from neighboring nodes.
>
> W2. Thanks for your valuable suggestions. (1) For example, in the first prompt of Appendix C, we use text summarization to generate the initial textual representations for each node, analogous to initializing dense vector embeddings in embedding-based GNNs. Building on this, as shown in the second prompt, we employ aggregation and update processes to guide the LLM in performing GNN-style message passing, moving beyond simple summarization. (2) Theoretically, at each layer, we emulate the loss function in GraphSAGE by employing prompts such as "Note connected nodes should share similar semantics and vice versa." These cumulative, layer-by-layer prompts are analogous to layer-wise loss combinations, formally akin to mean pooling across all GNN layer embeddings, and mathematically extend beyond the textual representation in the final layer. As demonstrated in [4][5], such structural information can be effectively preserved. Experimentally, in Table 9 of Appendix F.2, we present the final textual representations alongside the original keywords from selected papers (the keywords are not included in the prompts and serve as ground truth). The observed similarity demonstrates that our prompt-based GNN effectively captures the core semantics of these nodes.
>
> W3. Apologies for any misunderstanding. We would like to clarify that our work focuses exclusively on GFM for text-attributed graphs. In this framework, each node corresponds to a textual representation, referred to as a language-based ID, which is applicable and comparable across all text-attributed graphs and warrants the development of a universal vocabulary for such graphs. This vocabulary enables the construction of pure language instructions, which are crucial for the subsequent multi-instruction fine-tuning framework for LLM inference. Our approach differs from [1,2] and is not applicable to [3] (as indicated in the response above).

---

> ### Author Response · Authors · 2024-11-24
> **Response to  Reviewer p4FX (2)**
>
> W4. Thank you for your suggestions. (1) Our work represents an attempt at LLM as GNN, where we use prompts to guide LLMs in replicating message passing for each node, aligning with the fundamental principles of GNN-based models. This approach demonstrates how LLMs can function as GNNs, fundamentally differing from existing LLM for GNN and GNN for LLM models, where GNNs and LLMs operate independently. (2) Apologies for any confusion. In Figure 2, we compare the prompt-based GNN (our method) with the embedding-based GNN (e.g., GraphSAGE) to highlight the faithful replication of the GNN workflow. It is important to note that PromptGFM is not a direct combination of a GNN and an LLM. (3) Regarding the second question, PromptGFM extends beyond pure information propagation. We argue that the essential elements of a GNN include input, neighbor sampling, aggregation-update mechanisms, multi-layer message passing, and optimization. In PromptGFM, we design specific prompts to empower LLMs to faithfully replicate each of these elements in the GNN workflow. Therefore, the term "prompt-based GNN" effectively summarizes our approach.
>
> W5. Thanks for your comments. We will replace this term and thoroughly review the entire text for precise wording.
>
> W6. Thanks. We will summarize these sections and move some detailed explanations to the Appendix.
>
> Q1. Apologies for the confusion. The construction of the graph vocabulary is detailed in Section 4.2. Following a comprehensive analysis, we directly use the final textual representations as language-based IDs. These IDs are then indexed from the graph vocabulary and integrated into task-oriented prompt templates to create comprehensive instructions. Then, we fine-tune the LLM under a multi-instruction fine-tuning framework. We will consider reorganizing this content for better clarity.
>
> Q2. Sorry for misunderstanding. We prompt LLM to iteratively refine textual representations, which goes beyond simple rephrasing. During the multi-round prompting process with LLMs, both semantic fusion and structural information capture occur simultaneously. Using language-based IDs, we can construct readable instructions for LLM inference, resolving modality incompatibility and facilitating positive transfer.
>
> Q3. Thanks for your question. Both datasets do have text attributes from other sources, as indicated in the datasets provided in the code repository.

---

> > ### Comment · Reviewer_p4FX · 2024-11-25
> > **Response to authors**
> >
> > Thank you authors for addressing my questions and concerns.
> >
> > However, my main concerns about the scalability of the work remained because the method doesn't scale with large graphs even by sampling methods. Also, the generated text from a graph may exceed the eligible length LLMs allow for input. More importantly, the way LLMs do the message-passing process, based on the experimental results, is more like summarization than information propagation over the neighborhoods. Furthermore, I think authors should ensure that nodes are preserved through multiple layers of LLM up to the last layer, and it'd be helpful to show how they handle retrieving the final representation of the nodes.
> >
> > Therefore, I will keep my score as I think the paper is not ready to be published.

---

### Note · Authors · 2024-12-04

**Comment:**

I have read and agree with the venue's withdrawal policy on behalf of myself and my co-authors.

**Withdrawal Confirmation:**

I have read and agree with the venue's withdrawal policy on behalf of myself and my co-authors.